# Examining Perceived Stress and Coping Strategies of University Students during COVID-19: A Cross-Sectional Study in Jordan

**DOI:** 10.3390/ijerph19159154

**Published:** 2022-07-27

**Authors:** Fahd Alduais, Abeer Ihsan Samara, Heba Mustafa Al-Jalabneh, Ahmed Alduais, Hind Alfadda, Rasha Alaudan

**Affiliations:** 1Department of Accounting, Philadelphia University, Jarash Road, 20 KM, Amman 19392, Jordan or accofahd@hotmail.com (F.A.); asamara@philadelphia.edu.jo (A.I.S.); 2Department of Accounting, National Institute of Administrative Sciences, Ibb 70270, Yemen; 3Department of Business Administration, Philadelphia University, Jarash Road, 20 KM, Amman 19392, Jordan; hjalabneh@philadelphia.edu.jo; 4Department of Human Sciences (Psychology), University of Verona, 37129 Verona, Italy; 5Department of Curriculum and Instruction, King Saud University, Riyadh 11451, Saudi Arabia; halfadda@ksu.edu.sa

**Keywords:** perceived stress, coping strategies, higher education, university students, COVID-19, psychological well-being, Jordan

## Abstract

COVID-19 has changed our lives in all arenas, including higher education and psychological well-being. Three objectives were set forth in this study. We started by examining issues related to online education during the pandemic in Jordan, particularly for students pursuing business studies. Second, we assessed academic, behavioural, and financial stressors that business students experience at Jordanian higher education institutions. Lastly, we examined the possible coping methods students employed to cope and adapt during the pandemic. A descriptive cross-sectional study was conducted based on the perceived stress scale distributed to 473 Jordanian undergraduate students (18–27 years of age), across both public and private universities. Results showed an association between academic, behavioural, and financial stressors and students’ perceived stress. While students perceived various levels and types of stress during COVID-19, including academic, behavioural, and financial, they also experienced new online skills. However, despite coping with stress, some students (especially females) displayed more stress because of the deficient course content, which added to their perceived stress and left them feeling unmotivated. This study contributes to bettering the university students’ mental health.

## 1. Introduction

Colleges and institutions throughout the world, including Jordan, have been forced to close during the spread of COVID-19. As a result of the spread of COVID-19, students and lecturers who were unable to attend school have found that online learning, which offers education through digital platforms rather than in person, has revolutionised education [1,2]. Currently, the literature on online learning contains a great deal of conflicting information. Several recent studies find that online learning is gaining popularity and success, while others believe it is still difficult for students [3,4,5,6,7]. The COVID-19 pandemic has illustrated that the education system is highly vulnerable to external threats [8]. These differences are the result of students’ increasing comfort with online learning and their enrolment solely for convenience, rather than considering how they would best learn the material. With so many lecturers failing to switch to online learning and keeping all students engaged, the COVID-19 epidemic has been a considerable concern for many in higher education [9].

This study examined the following questions: What challenges did students encounter when switching from a face-to-face mode to a virtual classroom during the pandemic? How did students at Business colleges in Jordan engage in online learning? Does COVID-19 pose a threat to higher education? Can online learning be as effective as face-to-face training? What steps were taken by business students to cope with the pandemic? In what ways did students of business respond to the COVID-19 pandemic? Is there a way for higher education institutions to reduce such challenges to improve the quality of learning during the pandemic? To address the research questions, a cross-sectional study was conducted at Jordanian universities. For college and university authorities to effectively expose the aspects driving student engagement in online learning, they may need to provide suitable learning facilities. In addition to benefiting from the research’s results, online instructors may also gain insight into the study’s impact and the extent to which they manage their sessions to ensure that no student is left behind. 

The study examined business students’ perceived stress, online learning, and the academic, financial, and behavioural stressors related to COVID-19. Several recent studies have examined factors affecting students’ participation in online education [1], but more research is needed, especially with the emergence of many crises that may lead to the transformation of online education, such as recent epidemics that may affect students’ participation in online learning. In response, this study sought to bridge the knowledge gap by surveying business students at Jordanian universities regarding the variables impacting their participation in online learning. The study contributes to the existing body of literature on online learning by providing an understanding of the strategies employed by students in dealing with such pandemics and the challenges faced in combating them.

### 1.1. E-Learning, Stress and Coping Strategies

Learning institutions such as universities must increasingly use innovative methods of instruction to remain competitive, and adopting E-learning has become an unavoidable option in the COVID-19 era and beyond [10]. Therefore, introducing an e-learning portal will increase the possibility of using mobile devices such as smartphones and tablets to access lessons. Mobile devices have shown benefits for engaging students and teachers in teaching and self-regulated learning [11]. Business students face a serious problem with the COVID-19 pandemic, which has become a significant stressor and negatively impacts their academic achievements. In this study, we sought to determine whether the current pandemic impacts undergraduate business students’ learning. Moreover, we investigated the association between their stress level and coping strategies, educational, financial, and behavioural variables. Recently, the internet has been used as a platform for educational opportunities and has become increasingly important.

Consequently, it is essential to have reasonable expectations and understand how they can be utilised to assist students that the pandemic has impacted. In many nations, the higher education sector has resorted to various strategies and procedures to teach students and reduce the disruption caused to educational programs and academics. For example, all students can access educational channels on national television, video audio lessons on Facebook, Zoom, Microsoft Teams, and Skype, and textbooks can be digitised and made available online.

According to stress and coping theory [12], two processes, appraisal, and coping, play an essential role in the ongoing relationship between an individual and their environment. This framework can formulate and test hypotheses regarding the stress process and its relationship to physical and mental health [13]. Stress is a relationship between an individual and their environment perceived as personally significant, which exceeds an individual’s capacity to cope [14]. Stress appears in nearly all biological, social, and behavioural sciences literature. It is commonly found in economics, political science, business, and education [12]. Our study adopted the emotion-focused, which is directed toward changing one’s emotional reaction [12,15,16]. According to Weiten, Dunn, and Hammer [17], coping strategies represent reactions or attempts to cope with stress. However, Folkman [13] defines coping strategies as acts or methods of thinking used to cope with an uncomfortable or stressful situation. Coping strategies relate directly to how a stressor may affect an individual’s physical, psychological, and behavioural responses [18]. Stress-coping strategies aim to improve control perception and decrease momentary aversive attributes as a reaction to a stressful event. The pandemic has had a negative impact on education, and as a result, students’ routine life and psychological health could not remain unaffected [19]. Despite recent studies finding preventive behaviours effective, the effects of these coping strategies on well-being have not been studied [20,21,22], and the effects of them on psychological well-being are unknown.

### 1.2. Recent Evidence on Stress, Coping and Education during and aftter COVID-19

Perceived stress has no borders, and recent research reported the increase in perceived stress among higher education students regardless of their level. A study comparing data for students in China, Ireland, Malaysia, Taiwan, South Korea, the Netherlands, and the United States reported a negative association between perceived stress, anxiety and psychological resilience. The researchers proposed an increase in promoting psychological resilience among university students to increase psychological resilience and decrease stress and anxiety [23]. The perceived stress extends to include teachers in the higher education sectors as evidenced in a study on Portuguese teachers [24]. However, COVID-19 and workloads are not the only main causes for perceived stress among the higher education community (i.e., students, teachers, administrators); discrimination based on colour could be also another cause as reported on a study conducted in South Korea on black teachers [25].

Several factors could influence the perceived stress on students and teachers. For in-stance, a study in Spain reported that emotional intelligence and a feeling of achievement enhance coping strategies against perceived stress [26]. Another study in Spain empha-sized the need for further training to university students to help them manage perceived stress due to the pandemic [27]. One more study in Poland reported an increase in the level of educational burnout, lower life satisfaction and less-effective coping strategies for perceived stress during the quarantine of COVID-19 by Polish students [28]. Similarly, a study on higher education students in Ghana reported the impact of technostress on the students’ academic achievement and academic productivity [29]. Given this, a study con-ducted on students from seven countries (i.e., China, Ireland, Malaysia, South Korea, Tai-wan, the Netherlands, and the United States) emphasized the importance of sleep educa-tion and resilience training in the higher education sector to enhance coping strategies and decrease the perceived stress [30]. Above all these factors, diagnosis and measurement of the perceived stress during COVID-19 form the essential steps towards building stronger coping strategies among the higher education community [31].

Very recent research on perceived stress and COVID-19 reported that work changes and financial insecurity are also among the most common stressors in the higher education community in a study conducted on early care and education in the United States [32]. Among the other factors that could contribute to decreasing the perceived stress are perceived social support and hope, as examined on a sample of Chinese shadow teachers [33]. Furthermore, a study on student nurses in Poland found that a higher level of perceived stress corresponds to a lower level of self-efficacy, life satisfaction, life-orientation, and self-esteem of students [34]. Apart from that, another study reported that perceived stress was managed by administrators and managers in higher education using instrumental support, problem-focused strategies, religion, behavioural disengagement, and a sense of humour [35].

### 1.3. The Present Study

Based on the previous literature review, the purpose of this study was to use structural equation modelling to investigate the relationships among online learning perceptions, online course performance, and course satisfaction, by testing the hypotheses of the effect of stress factors and perceived stress on students’ academic achievements and future. A representation of the model tested in this study is illustrated in Figure 1.

Increasingly, face-to-face instruction was being replaced by online instruction. Students may experience emotional stress due to the lockdown, the feeling of home sickness, and the impending fear of mortality. As well as academic stress (such as the pressure of an extended curriculum or the stress of not being able to comprehend the desired knowledge in time), behaviour stress and financial stress (for example, the fear of not being able to pay the university’s financial instalments or that the family will be impacted financially by a pandemic), business students face significant behavioural issues as well. In this study, a conceptual framework (Figure 1) was developed that suggests that business students are subjected to a stressor during online learning due to the COVID-19 pandemic, and that the perceived stress level is determined by personal characteristics, attitudes, and behaviours, academic achievement, as well as financial capacity and learning environments. The phenomenon of PS (perceived stress) occurs because of an increase in pressures and a decrease in resources. To this end, we sought to determine: (1) the level and sources of stress experienced by business students in Jordanian universities owing to online learning; (2) the impact of perceived stress on students’ academic performance; and (3) identity and effectiveness of coping strategies among students. Focusing on the stressors and risk factors for PS among business students in such an epidemiological period is indispensable for developing conditioned and targeted interventions to maintain good mental health. In addition, this study is designed to enhance the impact of business education by identifying facilitators and barriers to online learning and creating innovative online coping strategies for business students during times of crisis.

Furthermore, this hypothetical model, derived from the literature review, was used to test the following hypotheses:

**H1.** 
*Stressors associated with COVID-19 (academic, financial, and behavioural stress) are positively related to students’ perceived stress.*


**H2.** 
*Well-being coping strategies of students affect perceived stress negatively.*


**H3.** 
*Stressors associated with COVID-19 (academic, financial, and behavioural stress) are positively related to a student’s future academic risk.*


**H4.** 
*Perceived stress is positively related to students’ future academic risk.*


**H5.** 
*Perceived stress is negatively related to students’ performance achievement.*


**H6.** 
*There are significant differences in perceived stress, academic stress, financial stress, behavioural stress, and coping strategies scores with different students’ gender.*


## 2. Method

### 2.1. Sample

The population of interest for this descriptive cross-sectional study consists of all Jordanian university students who were still enrolled, or recently graduated, in any higher education institution in Jordan in 2022, regardless of their educational level. The accessible population is Jordanian students in business and administration departments across Jordan (https://mohe.gov.jo/Default/En) (accessed on 1 June 2022). As the sampling frame, we utilized the official website of the Ministry of Higher Education and Scientific Research in Jordan, statistical reports produced by university deans, vice deans, chairs of departments, and accessible social media groups for university students in Jordan. The survey was randomly distributed and shared with these officials, groups, and colleagues for the purpose of collecting data. A total of 474 Jordanian business students were included in the final sample, and were distributed across universities, undergraduate study levels, and genders. Table 1 and Table 2 provide detailed descriptions of the sociodemographic characteristics of those included in the study. 

A total of 474 business students participated in the study, with males representing 48.5% of participants and females representing 51.5%. Most participants were students from Philadelphia University (24.268%), followed by University of Jordan (23%) students. Other universities had varying participation proportions in both public and private higher education sectors. The sample was evenly distributed among first-, second-, third- and fourth-year students (23.8%, 29.5%, 19.2%, 23.6%, respectively), and the rest of the sample for the recently graduated (3.8%). See Table 2 for further details. 

Although this study sample and its cross-sectional design were not intended for generalizability, the results of the study may serve as an initial indication of the stress and psychological health of students during this pandemic. The study explores the students’ coping strategies and compares stressors at different study levels of universities. Even though these outcomes will vary by context (even within the Arabian context) and factors such as socioeconomic development and educational development, among others, the existence of perceived stress with its various types and coping strategies should be universal.

### 2.2. Measures

A Perceived Stress Scale of 5 items adapted from Cohen, Kamarck, and Mermelstein [36] was administered to assess how much people perceive recent life events as stressful on a scale of 1 to 5, ranging from “never” to “very often”. An exhaustive literature review also informed how to achieve the objectives of the study related to academic stress (4 items), financial stress (4 items), behavioural stress (4 items), and coping variables (1 item), using adaptions from [19,37,38,39]. The participants were asked how well they dealt with disruptions in their daily lives caused by COVID-19 compared to others around them, along a scale of 1 to 5, ranging from “not well at all” to “extremely well”; they were also asked to rate course contents and practical courses (1 item for each one) adapted from [19,40,41], and rate the academic future (1 item) adapted from [41]. There were 20 items (See Appendix A) attached as a Appendix A. Furthermore, the final grade or grade point average is also usually used as a dependent variable in research, indicating the individual’s overall learning performance as a dependent variable in their studies. Our study adapted academic performance from [42,43,44,45].

Because of the COVID-19 pandemic, the educational system has shown itself to be vulnerable to external threats. The universities’ lockdowns that began in the spring of 2020 reduced instruction and learning time, which is known to hinder students’ performance, with varying effects on different student groups. Many studies have explored the underlying factors of online learning success and failure and have addressed the critical factors impacting learner satisfaction [40,46,47,48,49,50,51]. Previous studies have determined factors deemed to be influential on student performance or course satisfaction in online learning environments; however, little research has been performed to explore the students’ general online learning perceptions, performance, practical courses, and contents course satisfaction, and their impact directly on academic performance [40,42]. As a result, further analysis was performed to strengthen our findings using the course contents and practical lessons as variables related to grade point averages (hereafter GPA) and students’ performance, academic stress, financial stress, and behavioural stress.

Following the completion of the first draft of the survey, this was shared with the co-authors for evaluation against the study’s objectives; they completed a trial version of the survey and provided feedback on its readability and usability. There were both linguistic and technical modifications applied until an agreement was reached on the final version, although the survey content did not change. There were checks performed on both the Arabic and English versions, although only the Arabic version was used for data collection. More importantly, however, we conducted reliability analysis, and Cronbach’s alphas within the 25 items of the stress scale ranged from 0.68 to 0.82. Overall, the perceived stress scale was moderately reliable (25 items: α = 0.72). 

### 2.3. Design 

We conducted a cross-sectional study to examine perceived stress (academic, behavioural, and financial) and coping strategies employed by Jordanian undergraduate students to cope with this stress during COVID-19. Since everyone was exposed to this pandemic, and it was impossible to distinguish between people who were contaminated with COVID-19 and those who were not, an analytical cross-sectional study was not feasible. The study included quantitative analysis based on a survey where both descriptive and inferential analyses were conducted.

### 2.4. Procedure

A cross-sectional study was conducted among business students in Jordan in May and June 2022, using an online survey to gather information about online education outcomes. Participants—business students in Jordan—were invited to participate in the study via the official universities’ mailing systems and appropriate social media groups. They had to fill in an online survey available through Google Forms. The effectiveness of online learning was evaluated using a survey, which was designed and sent to participants via an online form developed and administered via Google online forms. In the survey, participants were asked to provide information about sociodemographic characteristics (age, gender, college attended, and academic year).

This paper aimed to find the general perspective of business students towards online learning during COVID-19. Therefore, we used the research instrument from the previous studies on perceived stress and online learning among Jordanian business students. We adopted the items from [4,15,19,36,37,38,39,52,53,54,55]. It is essential to verify the validity and reliability of the constructs and the scale [56], so the information gathered through the online survey was analysed based on the frequency of business students’ responses, which was expressed as a percentage. 

The demographic information was collected using a Likert scale. The results are presented as a percentage of business students’ responses, with organisation and statistical analysis performed using the Excel spreadsheet and statistical Stata software. Pearson’s linear correlation assesses the relationships between continuous variables, and the frequency distribution compares categorical variables. A one-way ANOVA test and independent sample *t*-tests were used to examine the differences between perceived stress, academic stress, financial stress, and behavioural stress for gender variables. Multiple linear regression analyses were performed to determine the predictors of perceived stress and academic achievement (GPA). A *p*-value of 0.05 was considered a cut-off for statistical significance for all purposes.

Finally, it is worth mentioning that all participants consented their voluntary participation in the study using an online consent form included in the first part of the online survey. This consent form and the procedure were approved by the ethical committee in the department of accounting, Philadelphia University, Jordan. 

## 3. Results

In this study, we assessed Jordanian undergraduate students’ perceived stress (academic, behavioural, and financial) regarding their online education experiences during the COVID-19 and the coping strategies they used to cope with stressors. The data are presented below in four sections. We begin by examining the demographic characteristics of the population. Secondly, we introduce descriptive and correlational analyses to examine associations and differences between the different measured stressors and coping strategies based on gender and academic level. Our third analysis focuses on testing our six hypotheses. Finally, we review findings that emerged from the three previous analyses regarding course content and practical lessons, as well as their relevance to academic achievement and perceived stress.

### 3.1. Use of Online and Virtual Learning Tools

Data showed that participants used several electronic devices to study online. The most common device was the smartphone (62.87%), followed by laptop (20.25%), tablet (14.56%), and PC (2.32%) (Figure 2). The hours spent learning online ranged from 1 h/day to 10 h/day with an average of 5.32 h/day (Figure 3). Regarding the frequency of online studying hours, about 33.12% of participants spent up to 5 h/day in online learning, while 27.43% of participants spent 6 h/day, 17.09% of participants spent 4 h/day, and other hours/day had varying proportions of participation (Table 3).

About (77.22%) of participants evaluated the online course contents with 1–4 of 10 points (Figure 4), while (78.48%) of participants evaluated the online learning in practical lessons with 1–4 of 10 points (Figure 5). About (86.7%) of participations rated the risk of academic future due to COVID-19 as very high, with 6–10 of 10 points (Figure 6).

Participants showed that the online study materials were available mostly through university platforms (61.6%), online classes (32.7%) and YouTube videos (2.5%), followed by educational applications (1.5%), pdf lectures (0.8%), educational websites (0.6%) and e-books (0.2%) (Table 2). Different online tools have been used to access online classes. The distribution of these online tools was as follows: Microsoft Teams (95.6%), Zoom (3.0), and Skype (0.2) (Table 3).

Table 4 compares the means of demographic variables and perceived stress scores. In perceived stress, financial and behaviour stress, females show a significantly greater number of 3.92, 3.67, and 3.57, respectively; except in the academic stress, the males show a significantly greater number of 3.51.

### 3.2. Descriptive and Correlation Analysis

In Table 5, the means for perceived stress and stressors are perceived stress 3.86, academic stress 3.41, financial stress 3.56, and behavioural stress 3.45. These are very high, so stress increased during the pandemic, affecting their academic, financial, and behavioural situations. The students showed concerns about the content of both the lessons (3.60) and the practical lessons (3.49) during the pandemic. Overall, students are concerned about the quality of the contents and the practical lessons. Furthermore, participants were concerned that COVID-19 would adversely affect their future academic achievement (M: 7.795).

In Table 6, according to the Pearson correlation coefficient, there is a significant positive correlation (0.512 **, *p*-value <0.01) between the PS and the impact of COVID-19 on students’ future academic performance. Additionally, there is a positive correlation between AS (0.543 **, *p*-value <0.01), FS (0.253 **, *p*-value <0.01) and BS (0.391 **, *p*-value <0.01) and PS, and a negative correlation between the coping strategies (−0.179 **, *p*-value <0.01) and PS during the pandemic.

### 3.3. Hypotheses Test

In Table 7, regarding sources of stress, academic stressor (β = 0.395 ***), financial stressor (fs, β = 0.112 ***) and behavioural stressor (bs, β = 0.348 ***) increased students’ PS. The stressors positively affect PS, which confirms the correlation in Table 6. Accordingly, the findings of this module regression support hypothesis H1. The persistence, attitude, and flexibility of students’ coping strategies are shown in Table 6. Based on students’ responses, they used to deal with stress moderately well, with high responses for 130 males (54.17%) and 110 females (45.83%); the other responses are differentiated. However, in Table 7, the second regression model between coping strategies and PS reported the coefficient (CS, β = −0.112 ***), which played a significant negative role in PS. Our findings support H2, which means the students applied coping strategies during the pandemic, decreasing their PS. However, most of the participants with slight and moderate well coping strategies levels (Table 7) show a significantly greater number of (*n* = 141) and (*n* = 240), respectively, and show the female greater number (*n* = 97 + 110), *p*-value < 0.05%.

The stressors also impact the academic future; the regression in model 3 explains this relationship and supports hypothesis H3. The PS positively and significantly affected academic achievement risk in the future (β = 1.562 ***), which supports hypothesis H4. However, the PS has a negative impact on academic achievement (β = −0.095 **), and the findings confirm our H5. See Table 8 for further detail. 

Moreover, to test hypothesis H6, an Independent *T*-test was performed. Our findings indicated statistically significant differences in PS, AS, FS, BS, and CS scores with different students’ genders (*p* < 0.05), see Table 9.

### 3.4. Further Analysis

The participants evaluated the course contents and practical lesson as “very bad” (Figure 4 and Figure 5). Table 10 shows the robustness of the findings; PL is positively related to academic performance (β = 0.103 ***). They believe the PL is a significant factor in enhancing their academic achievement. However, the course contents negatively affected the GPA (β = −0.053 **), where they believed that even if the contents increased during the lockdown, the performance decreased as well. Due to the pandemic, academic performance was negatively affected, including course content and practical lessons. As a result, poor distance education was provided, increasing student stress levels. Academic institutions should invest more in developing their curriculum and employ effective instruction methods and strategies [19], so that online learning will be more sustainable while instructional activities will become more effective [8].

## 4. Discussion

The purpose of this study was to examine the perceived academic, behavioural, and financial stress of university students in Jordan. The sample included 474 Jordanian undergraduates from both public and private universities throughout Jordan. There were four key findings of this study. First, university students engaged in higher levels of technology usage in response to the pandemic, and preliminary results indicate that females reported higher levels of financial and behavioural stress than males, who showed higher academic stress. The correlational analysis indicates that perceived stress is associated with future academic risk and is positively correlated with academic stress, behavioural stress, and financial stress. Consequently, the pandemic increased the level of stress, causing academic, behavioural, and financial pressure to increase. Second, coping strategies were negatively associated with perceived stress, which in one way or another helped the students to reduce and resist perceived stress. Third, the regression analysis provided evidence supporting the six alternative hypotheses. Among these are the findings that academic, financial, and behavioural stressors have a positive effect on students’ perception of stress and their future academic risk. Additionally, students’ coping strategies are negatively associated with their perceived stress and academic performance. Apart from that, perceived stress is positively related to a student’s future academic performance, and there are statistically significant differences based on perceived academic, behavioural, and financial stressors, as well as coping strategies. In Figure 7 below, we demonstrate these in our proposed model. Fourth, the Jordanian undergraduate students surveyed negatively viewed the course content and practical lessons. The data analysis indicated a positive correlation between practical lessons and academic achievement, as opposed to a negative correlation between course content and students’ GPA. 

The four patterns of results are consistent with previous literature that shows COVID-19 lockdown adversely affects academic performance in most of the participants. Online education has the advantage of allowing students to perform self-study independently and learn new techniques [57]. However, the primary challenge lies in the content and practical lessons, and this is consistent with other studies [19,39]. An online education system may not be sufficient for fulfilling a student’s business competencies. In online education, several improvements can be made, including making it more interactive [58], showing business processes in real-life situations, and improving lesson content. Our findings appear to confirm the importance of face-to-face education. Further, coping strategies increase resilience and help in reducing stress levels as reported in several previous studies in other contexts including: Portugal [24], South Korea [25], Poland [28], and China, Malaysia, Taiwan, South Korea, the Netherlands, and the United States [23,30]. Absence or reduced interaction with students, absence of nonverbal communication, inability to teach face-to-face, and inability to check students’ understanding of the material are frequent topics in the literature on distance education [58,59,60]. More importantly, our results highlighted the importance of training to increase students’ coping strategies and help them manage perceived stress, which is consistent with findings in other contexts reporting the promotion of psychoeducational strategies in Peruvian higher education institutions [61]. 

## 5. Conclusions

In this study, we investigated whether the pandemic affected undergraduate business students’ learning, stress levels, coping strategies, and psychological variables. As a result of the COVID-19 pandemic lockdown, students’ academic performance was impacted. Evidence from this study strengthened the conclusion that the students’ perceived stress increases their academic, behavioural, and financial stress. Together, they adversely affected their academic achievement, which was hampered by the poor course content and practical lessons. These findings suggest a need to improve the technological infrastructure of Jordan’s education system to ensure more efficient and effective use of online education during the pandemic or similar situations. This should include a greater focus on training university teachers in online teaching, curriculum development, and instructional development at all higher education institutions in Jordan. Even though students managed to maintain the flow of education through several technological means, there was also psychological pressure that could be alleviated if a better response were made to the pandemic in higher education. 

## 6. Limitations

Certain limitations of this study could be addressed in future research. For instance, the study sample was limited to undergraduate students majoring in business and administration studies. We made this decision due to the inaccessibility of other students, requiring lengthy bureaucratic processes to reach them. As another example, the level of the students in this study was limited to undergraduate students, including recent graduates. While the study planned to include graduates and even postdoctoral fellows, only a few responses were received from graduate students, and these were removed from the sample to preserve the sample’s internal validity. The third limitation of this study is the comparison of perceived stress based on university type (public/private), university ranking (among the top 500/after the top 500), or socioeconomic status (developed/developing; city/town/country). The data collected included all these variables; however, the returned data was insufficient to carry out comparisons among them. One more limitation is that the measure used was a self-reporting tool which means that these results might change if using other methodological techniques such as repeated measure or formal assessment of perceived stress. 

## Figures and Tables

**Figure 1 ijerph-19-09154-f001:**
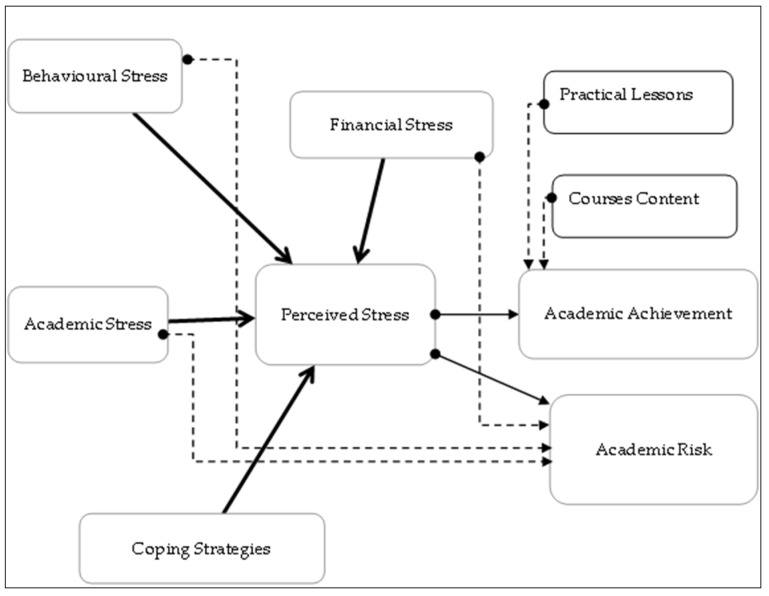
The theoretical structural model.

**Figure 2 ijerph-19-09154-f002:**
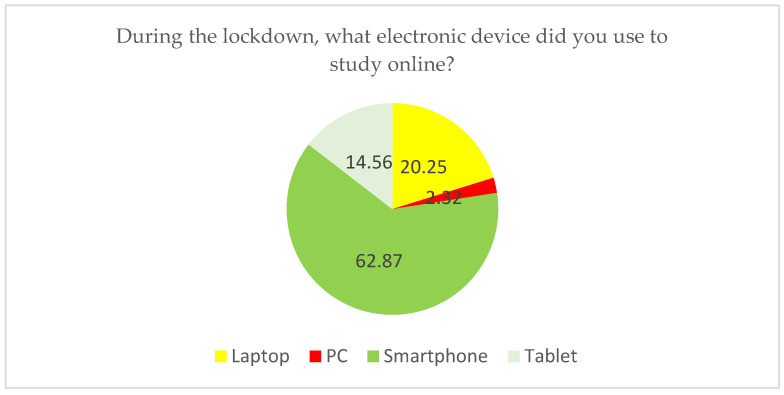
The device used by participants to access online.

**Figure 3 ijerph-19-09154-f003:**
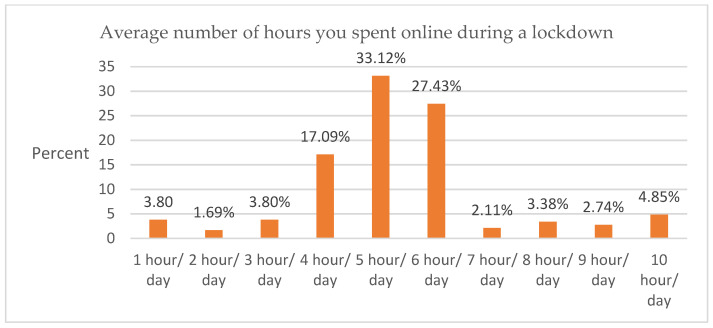
Average number of hours you spent online during a lockdown.

**Figure 4 ijerph-19-09154-f004:**
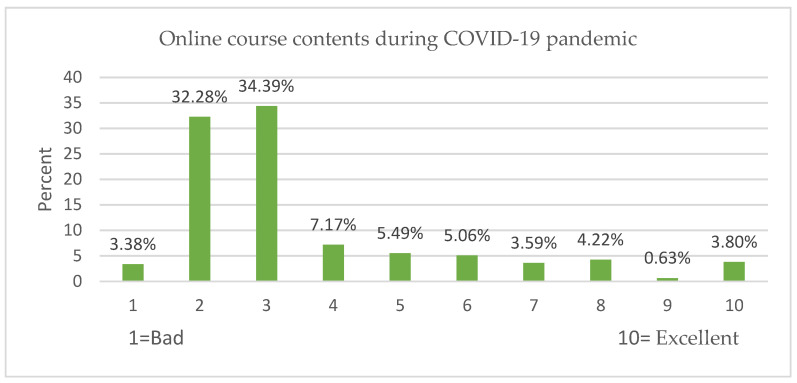
Evaluation of online course contents during lockdown.

**Figure 5 ijerph-19-09154-f005:**
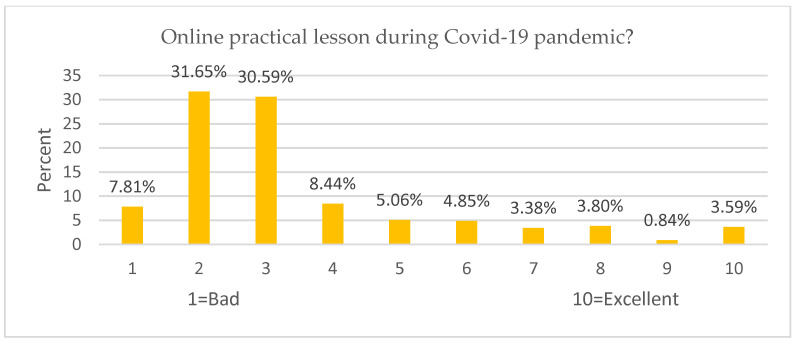
Evaluation of online course contents during lockdown.

**Figure 6 ijerph-19-09154-f006:**
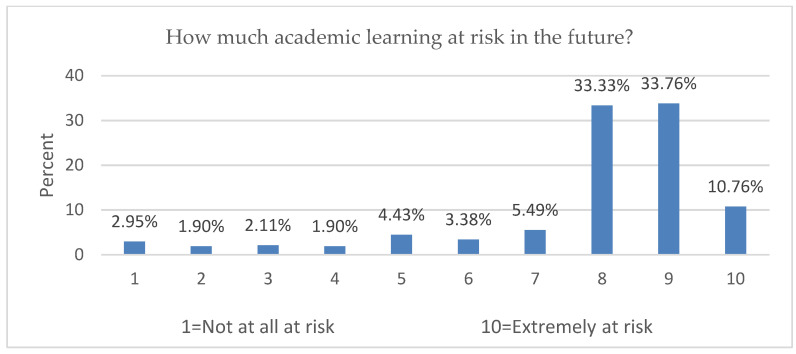
Evaluation of the impact of COVID-19 on academics in the future.

**Figure 7 ijerph-19-09154-f007:**
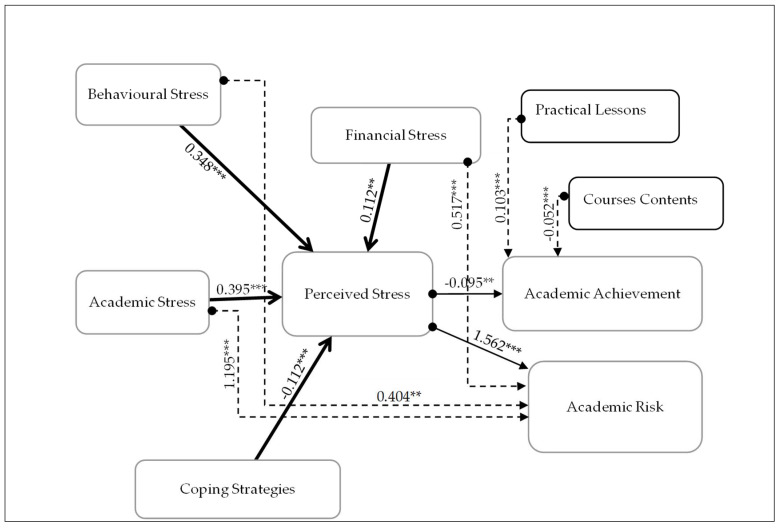
Final theoretical framework. *** *p* < 0.01, ** *p* < 0.05.

**Table 1 ijerph-19-09154-t001:** Sample description.

Description	Responses
Total received responses	495
Mismatched GPA *	−12
Missed University name.	−5
post-graduate	−4
Final sample	474

* GPA: Grade point average.

**Table 2 ijerph-19-09154-t002:** Participants characteristics.

Socio-Academic Characteristics	Total (%)
**Age range, *n* (%)**	
18–20	136 (28.7)
21–23	227 (47.9)
24–26	101 (21.3)
≥27	10 (2.1)
**Gender**	
Male	230 (48.5)
Female	244 (51.5)
**Academic year**	
1st year	113 (23.8)
2nd year	140 (29.5)
3rd year	91 (19.2)
4th year	112 (23.6)
Recently graduated	18 (3.8)

**Table 3 ijerph-19-09154-t003:** GPA and used online tool.

Socio-Academic Characteristics	Total (%)
**GPA**	
<2.00	3 (0.6)
2.00–2.5	154 (32.5)
2.6–3.50	224 (47.3)
3.6–4.0	93 (19.6)
**Use of online learning tool**	
Zoom	14 (3.0)
Microsoft Teams	453 (95.6)
Skype	1 (0.2)
Others	6 (1.3)
**Use of virtual learning tools**	
University platforms	292 (61.6)
Online classes	155 (32.7)
Educational websites	3 (0.6)
YouTube videos	12 (2.5)
E. Books	1 (0.2)
Educational application	7 (1.5)
PDF lectures	4 (0.8)

**Table 4 ijerph-19-09154-t004:** Perceived stress and stressors score by gender.

		Perceived Stress		Academic		Financial		Behavioural	
		M	%	M	%	M	%	M	%
Gender	M	3.79	48.5%	3.51	48.5%	3.45	48.5%	3.33	48.5%
	F	3.92	51.5%	3.32	51.5%	3.67	51.5%	3.57	51.5%

**Table 5 ijerph-19-09154-t005:** Descriptive statistics.

						Shapiro–Wilk (Normality)	
	N	M	SD	Minimum	Maximum	W	*p*
av_hours	474	5.32	5	1.82	1	10	0.896
PS	474	3.86	0.712	1	5	0.741	<0.001
AS	474	3.41	0.712	1	5	0.882	<0.001
FS	474	3.56	0.659	1	5	0.814	<0.001
BS	474	3.45	0.591	1	5	0.877	<0.001
CS	474	2.64	0.842	1	5	0.861	<0.001
RAF	474	7.8	2.07	1	10	0.761	<0.001
CC	474	3.60	2.138	1	10	0.783	<0.001
PL	474	3.49	2.163	1	10	0.804	<0.001

Abbreviations: PS (Perceived Stress), AS (Academic Stress), FS (Financial Stress), BS (Behavioural Stress), CS (Coping Strategies), RAF (Risk at Future), CC (Course Content), and PL (Practical Lessons).

**Table 6 ijerph-19-09154-t006:** Correlation analysis.

	Gender	Age	Level	GPA	PL	CC	RAF	PS	AS	FS	BS	CS
Gender	1.000											
Age	0.058	1.000										
Level	−0.007	0.758 **	1.000									
GPA	0.206 **	0.437 **	0.487 **	1.000								
PL	−0.077	0.202 **	0.362 **	0.238 **	1.000							
CC	0.036	−0.184 **	−0.110 *	−0.060	0.419 **	1.000						
RAF	0.002	−0.112 *	−0.189 **	−0.040	−0.329 **	−0.439 **	1.000					
PS	−0.031	−0.146 **	−0.264 **	−0.122 **	−0.424 **	−0.430 **	0.512 **	1.000				
AS	−0.284 **	−0.279 **	−0.220 **	−0.106 *	−0.328 **	−0.426 **	0.519 **	0.543 **	1.000			
FS	0.168 **	−0.016	−0.148 **	−0.081	−0.454 **	−0.202 **	0.335 **	0.253 **	0.338 **	1.000		
BS	0.220 **	−0.069	−0.123 **	−0.180 **	−0.408 **	−0.278 **	0.226 **	0.391 **	0.315 **	0.312 **	1.000	
CS	0.044	−0.035	0.000	−0.015	−0.111 *	0.147 **	0.077	−0.179 **	0.098 *	0.167 **	0.056	1.000

** Significant at the 0.01 level (2-tailed). * Significant at the 0.05 level (2-tailed).

**Table 7 ijerph-19-09154-t007:** Emotional well-being coping strategies of gender variable.

Gender	Not Well at All *n* (%)	Slightly Well *n* (%)	Moderately Well *n* (%)	Very Well *n* (%)	Extremely Well *n* (%)	Chi-Square (*p*-Value)
M	39 (88.64%)	44 (31.21%)	130 (54.17%)	13 (33.33%)	4 (40.00%)	52.227 (0.000)
F	5 (11.36%)	97 (68.79%)	110 (45.83%)	26 (66.67%)	6 (60.00%)	

**Table 8 ijerph-19-09154-t008:** Regression analysis Linear Regression Model Predicting Students’ Perceived Stress.

Module	(1)	(2)	(3)	(4)	(5)
Variables	PS	PS	RAF	RAF	GPA
AS	0.395 ***		1.195 ***		
	(0.044)		(0.138)		
FS	0.112 **		0.517 ***		
	(0.047)		(0.148)		
BS	0.348 ***		0.404 **		
	(0.053)		(0.167)		
CS		−0.112 ***			
		(0.039)			
PS				1.562 ***	−0.095 **
				(0.113)	(0.047)
Constant	0.905 ***	4.153 ***	0.481	1.771 ***	3.225 ***
	(0.159)	(0.107)	(0503)	(0.443)	(0.183)
Observations	474	474	474	474	474
R	0.676	0.132	0.597	0.537	0.093
R^2^	0.457	0.018	0.357	0.289	0.009

Note: module 1 tests H_1_: PS = β_0_ + β_1_AS + β_2_FS + β_3_BS + ε, module 2 tests H_2_: CS = β_0_ + β_1_PS + ε, module 3 test H_3_: ARF = β_0_ + β_1_AS + β_2_FS + β_3_BS + ε, module 4 test H_4_: ARF = β_0_ + β_1_PS + ε, module 5 tests H_5_: GPA = β_0_ + β_1_PS + ε. Standard errors in parentheses *** *p* < 0.01, ** *p* < 0.05.

**Table 9 ijerph-19-09154-t009:** Differences between groups.

Variables	Gender	N	M	SD	Sig. (2-Tailed)	t
PS	M	230	3.7904	0.81562	0.048	−1.981
	F	244	3.9197	0.59305		
AS	M	230	3.5120	0.82775	0.003	−3.592
	F	244	3.3186	0.56896		
FS	M	230	3.4522	0.70082	0.000	−4.369
	F	244	3.6670	0.59966		
BS	M	230	3.3337	0.62761	0.000	3.523
	F	244	3.5666	0.53134		
CS	M	230	2.5609	0.89795	0.043	−2.028
	F	244	2.7172	0.77907		

**Table 10 ijerph-19-09154-t010:** Further analysis of course contents and practical lessons and their impact on GPA.

Module	(1)	(2)
Variables	GPA	GPA
PL	0.103 ***	
	(0.026)	
CC		−0.053 **
		(0.027)
Constant	2.648 ***	2.743 ***
	(0.000)	(0.65)
Observations	474	474
R	0.180	0.094
R^2^	0.032	0.009

Note: module 1 tests GPA = β_0_ + β_1_PL + ε and module 2 tests GPA = β_0_ + β_1_CC + ε Standard errors in parentheses *** *p* < 0.01, ** *p* < 0.05.

## Data Availability

The data presented in this study are available on request from the first author. The data are not publicly available due to ethical restrictions.

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
