# Peer review of "Examining Perceived Stress and Coping Strategies of University Students during COVID-19: A Cross-Sectional Study in Jordan"

_ijerph, 2022, doi:10.3390/ijerph19159154_

Round 1
Reviewer 1 Report
Thank you for letting me review your work. Reading has been a pleasure. I found it very interesting. I will now proceed to the evaluation of it.
I consider this work to be original, novel and of clear application in the educational field. The issue under investigation is well defined. It seems to be an under-researched and useful topic.
The introduction is well organized and the studies cited are current. But I think it should be completed with studies that talk about stress during COVID-9, and if there are none, with stress during catastrophes or crises. The same is true for the lack of substantiation on the relationship between stress, education and coping, in order to adequately address the research objectives.
I think there is a typo because in the abstract it is indicated that the study is with students pursuing business studies and on page 2, line 85: In this study, we sought to determine whether the current pandemic impacts undergraduate medical students' learning, it is indicated that it is with medical students. Could you make a clarification in this sense?
The theoretical structural model on which it is based is not clear and cannot be understood with the diagram alone.
Method:
- The description of the sample section 3.1. should not be included in the results if not in the participants section, otherwise when you read that section you do not get an idea of what sample it has worked with. Since financial stress is a variable, I think it would be essential to know the socioeconomic level of the participants.
- Specify the acronym, for example GPA, and include what it means the first time it is named.
- Instruments: it is not possible to mix literature review with instruments, separate both aspects. In this section include only the instruments, number of items, an example of the type of questions and their psychometric properties for the sample. Indicate that it is an ad hoc instrument.
- Complete the study design indicating that it is quasi-experimental and quantitative. And as I see it is not only descriptive but also inferential, redo the explanation of the design. Figure 2 for me does not contribute anything, I would eliminate it.
- It is repeated at least 3 times that it is a descriptive cross-sectional design, leave only in the corresponding section that is the design and eliminate the rest.
- Separate from the procedure the analyses performed, that usually go together with the design. Has the effect size d and eta squared been calculated? If not, it should be completed and included.
- The procedure should include whether or not the study was approved by an ethics committee and if not approved indicate why legislation in your country was not necessary. Was a virtual informed consent signed, were there instructions, how long did it take to complete the questionnaire, over what period was the data collected?
Results:
- Separate sample characterization data, and take participants.
- Check if all graphs are necessary. E.g. participating universities I see it unnecessary, better to indicate how many universities and how many public and how many private.
- Include effect sizes to assess the magnitude of differences found.
Discussion/conclusions:
- It is very poor does not compare with other studies. Although it is a new study, there are studies on stress, strategies and so on in other times of crisis/disaster.
- Examples of stress and covid article, not from the Jordanian context but will be useful for comparison: https://www.revistaavft.com/images/revistas/2021/avft_1_2021/16_estres_academico_estudiantes.pdf
https://revistas.unheval.edu.pe/index.php/repis/article/view/867
https://ciencialatina.org/index.php/cienciala/article/view/238
- Regarding the representativeness of the sample, is it and if it is not include it as a limitation?
- The type of measure being self-report is also good to indicate it in limitations.
I hope you will take my recommendations into consideration, I believe they will help to improve the study.
Author Response
"Please see the attachment."

Reviewer 2 Report
I felt this was an overall logical presentation of the study. Hypotheses/research questions were answered, results presented for each concept along with detailed discussion. Conclusions, recommendations, and limitations were covered with suggestions for further research studies.
Introduction covered a succinct background of the elements/concepts for this study. The fact that you delineated the research questions and hypotheses was much appreciated. Many times, those are hard to find in an article. Kudos.
Line 3 & 26 and others that refer to the cross-sectional study. You discuss two types, should descriptive be included in the title?
Line 33-35: last line in abstract could be cleaned up for clarity.
The following lines and others in the article indicate brackets for a referenced article. It is difficult to know without searching the references if this indication is accurate. This may be a style/format I am not familiar with as to how an article is formatted. Line 105: According to [17]…..Line 106: However, [13] defines…..Line 161: A Perceived Stress Scale (5 items) adapted from [23] ….
Table 2: was the population age range enough for your study? 18 to ≥ 27? How much farther past 27 did you go?
Line 352 & 353: should these references be numbered and bracketed like all the others?
The link to supplementary material did not contain figures, tables, or videos. The only thing connected was the survey.
Overall, I found the article well thought out and presented. Minor grammatical and spelling errors suggest running through a Grammarly checker for help.
Author Response
"Please see the attachment."

Round 2
Reviewer 1 Report
Dear Authors,
thank you for taking into consideration most of my recommendations. And for arguing those that are not undertaken in a coherent manner.
I believe you have made a great effort to complete and improve the manuscript.
Best regards,
The reviewer.